# Polyphenols with Anti-Inflammatory Properties: Synthesis and Biological Activity of Novel Curcumin Derivatives

**DOI:** 10.3390/ijms24043691

**Published:** 2023-02-12

**Authors:** Yisett González, Randy Mojica-Flores, Dilan Moreno-Labrador, Luis Cubilla-Rios, K. S. Jagannatha Rao, Patricia L. Fernández, Oleg V. Larionov, Johant Lakey-Beitia

**Affiliations:** 1Center for Molecular and Cellular Biology of Diseases, Instituto de Investigaciones Científicas y Servicios de Alta Tecnología (INDICASAT AIP), Clayton, City of Knowledge, Panama City 0843-01103, Panama; 2Sistema Nacional de Investigación (SNI), SENACYT, Panama City 0816-02852, Panama; 3Center for Biodiversity and Drug Discovery, Instituto de Investigaciones Científicas y Servicios de Alta Tecnología (INDICASAT AIP), Clayton, City of Knowledge, Panama City 0843-01103, Panama; 4Laboratory of Tropical Bioorganic Chemistry, Faculty of Natural, Exact Sciences and Technology, University of Panama, Panama City 0824-03366, Panama; 5Center for Neuroscience, Instituto de Investigaciones Científicas y Servicios de Alta Tecnología (INDICASAT AIP), Clayton, City of Knowledge, Panama City 0843-01103, Panama; 6Department of Biotechnology, Koneru Lakshmaiah Education Foundation (KLEF) Deemed to be University, Vaddeswaram 522 302, India; 7Department of Chemistry, The University of Texas at San Antonio, San Antonio, TX 78249, USA

**Keywords:** curcumin, curcumin derivatives, succinate, difunctionalized, monofunctionalized, cytokine, IL-6, PGE_2_, anti-inflammatory activity, structure–activity relationship

## Abstract

Herein, we describe the synthesis and evaluation of anti-inflammatory activities of new curcumin derivatives. The thirteen curcumin derivatives were synthesized by Steglich esterification on one or both of the phenolic rings of curcumin with the aim of providing improved anti-inflammatory activity. Monofunctionalized compounds showed better bioactivity than the difunctionalized derivatives in terms of inhibiting IL-6 production, and known compound **2** presented the highest activity. Additionally, this compound showed strong activity against PGE_2_. Structure–activity relationship studies were carried out for both IL-6 and PGE_2_, and it was found that the activity of this series of compounds increases when a free hydroxyl group or aromatic ligands are present on the curcumin ring and a linker moiety is absent. Compound **2** remained the highest activity in modulating IL-6 production and showed strong activity against PGE_2_ synthesis.

## 1. Introduction

Polyphenols are natural products with several benefits for human health [1,2,3,4]. Polyphenols consist of a wide range of compounds that have multiple therapeutic properties [1,2,3,4]. Curcumin (**1**) is a fascinating polyphenol with several biological effects, such as anti-inflammatory, neuroprotective, antioxidant, and anticancer activities [2,5,6,7,8,9,10,11,12]. This pleiotropic molecule is found in the rhizome of *Curcuma longa*, and is a diarylheptanoid with two O-methoxyphenols attached to a β-diketone moiety connected by two symmetrical olefinic bonds [13]. However, curcumin has low bioavailability due to its low solubility in water and instability at physiological pH, and transforms into ferulic acid, vanillin, dehydrozingerone, and curcumin glucuronide [1,14,15,16,17]. Curcumin displays anti-inflammatory effects by modulating several pathways involved in the inflammatory process. This polyphenol inhibits the production of proinflammatory cytokines, such as tumor necrosis factor (TNF)-α and interleukins (ILs) 1, 2, 6, 8, and 12 [18], and regulates the activity of cyclooxygenase 2 (COX-2) [19]. Previous studies have shown that the anti-inflammatory effect induced by curcumin in vitro is achieved by regulating the activation of transcription factors such as activating protein-1 (AP1) and nuclear factor (NF) κB [18,19,20]. Curcumin inhibits NFκB by blocking IκB protein phosphorylation [18,19,20]. The curcumin-mediated downregulation of these intracellular signals leads to a reduction in the expression of cytokines [21]. It has been suggested that curcumin could be used as a nonsteroidal anti-inflammatory drug (NSAID) [22]. However, its low bioavailability due to its susceptibility to degradation in biological systems and poor solubility in water and plasma has prevented the medical use of curcumin [23].

Protecting the reactive sites of curcumin (the aromatic rings and the keto-enol region) (Figure 1) through the formation of derivatives could be an alternative strategy to improve its stability and take advantage of the benefits of this polyphenol [1,16,17]. Previous studies have shown that the phenolic rings and β-diketone moieties in the curcumin structure suffer from degradation by oxidation and hydrolysis [5,24]. Thus, protection of the hydroxyl groups might increase the bioavailability of this compound by suppressing degradation [22,25,26]. In this sense, some research groups have incorporated amino acids [27], glucose [28], alkyl [7] and succinyl groups into the curcumin structure [5]. In 2011, Wichitnithad et al. demonstrated that succinylation of curcuminoids protects the curcumin from hydrolysis and is an effective strategy against colon cancer [5]. Hence, we added the succinyl group to curcumin structure to protect the new molecules from degradation and to improve their bioavailability. 

In this research, new curcumin derivatives were synthesized to establish a structure–activity relationship (SAR) between curcumin and its derivatives [19]. Specifically, we focus on protecting the hydroxyl groups of the aromatic ring of curcumin through the incorporation of the succinyl group. We evaluated how the anti-inflammatory activity of the derivatives was altered as a result of the structural changes compared with curcumin.

## 2. Results

### 2.1. Synthesis of Novel Curcumin Derivatives

The synthesis of the curcumin derivatives was carried out in two stages. In the first stage, the alkyl succinate derivative was synthetized through the interaction of an alcohol of interest with the succinic anhydride (Appendix A). In the second stage, each alkyl succinate derivative (**S1***–***S9**) was coupled with curcumin to produce the curcumin derivative (Appendix A). Thirteen novel curcumin derivatives (**3***–***15**) were synthesized to determine their biological activity and establish a structure–activity relationship (SAR). 

Compound **1** was obtained commercially from Alfa Aesar with a 95% total curcuminoid content from turmeric rhizome. In this study, compound **1** was used as a reference to compare its activity to that of the novel curcumin derivatives. It is thought that the modification of curcumin, in addition to protecting its reactive sites (Figure 1), provides improvement in activity. We previously described a curcumin derivative, compound **2** (Figure 2), with a prominent inhibitory effect on IL-6 production [16]. Compound **2** has a different structural modification than the novel curcumin derivatives presented herein. Thus, this compound was also used as a reference to evaluate which modifications are more relevant to the anti-inflammatory activity of curcumin derivatives.

Compounds **S1**–**S9** were synthesized via a succinylation reaction (Appendix A). The reaction started with the interaction between adamantan-2-ol and DIPEA in pyridine (Py). After, succinic anhydride (SA) and 4-dimethylaminopyridine (DMAP) were added to give 4-((adamantan-2-yl)oxy)-4-oxobutanoic acid (**S1**, 86%). The same outcome was observed when 2-(hydroxymethyl)anthracene-9,10-dione and DIPEA in dichloromethane (DCM) reacted with SA and DMAP to give 4-((9,10-dioxo-9,10-dihydroanthracen-2-yl)methoxy)-4-oxobutanoic acid (**S2**, 89%). The same reaction occurred with diphenylmethanol to generate 4-(benzhydryloxy)-4-oxobutanoic acid (**S3**, 88%). Similarly, cyclohexyl(phenyl)methanol produced 4-(cyclohexyl(phenyl)methoxy)-4-oxobutanoic acid (**S4**, 98%). On the other hand, when 9H-fluoren-9-ol with DIPEA in Py was added to SA and DMAP, 4-((9H-fluoren-9-yl)oxy)-4-oxobutanoic acid **(S5**, 77%) was formed. When a solution of 2,3-dihydro-1H-inden-2-ol and DIPEA in DCM was added to SA and DMAP, the reaction produced 4-((2,3-dihydro-1H-inden-2-yl)oxy)-4-oxobutanoic acid (**S6**, 94%). Similarly, when a solution of (1R,2S,5R)-(-)-menthol and DIPEA in DCM was added to SA and DMAP, 4-(((1R,2S,5R)-2-isopropyl-5-methylcyclohexyl)oxy)-4-oxobutanoic acid (**S7**, 94%) was produced. In the same conditions, methanol generated 4-methoxy-4-oxobutanoic acid (**S8**, 67%). Finally, the addition of a solution of 1,2,3,4-tetrahydronaphthalen-1-ol and DIPEA in DCM to SA and DMAP afforded 4-oxo-4-((1,2,3,4-tetrahydronaphthalen-1-yl)oxy)butanoic acid (**S9**, 84%).

Compounds **3**–**15** (Figure 2) were synthesized by Steglich esterification. The reaction between curcumin and 4-(((1*R*,3 *R*,5 *R*,7*R*)-adamantan-2-yl)oxy)-4-oxobutanoic acid (**S1**) in the presence of DMAP, N-(3-Dimethylaminopropyl)-N′-ethylcarbodiimide hydrochloride (EDC), and DCM produced the difunctionalized (**3**, 37%), and monofunctionalized (**4**, 62%) succinate analogs. However, the reaction between curcumin and 4-((9,10-dioxo-9,10-dihydroanthracen-2-yl)methoxy)-4-oxobutanoic acid (**S2**) in the presence of DMAP, EDC, and DCM produced a monofunctionalized (**5**, 13%) succinate analog (Figure 2). The same outcome was observed when curcumin and 4-(benzhydryloxy)-4-oxobutanoic acid **(S3)** were reacted with DMAP, EDC, and DCM, which generated a monofunctionalized (**6**, 14%) succinate analog (Figure 2). The reaction between curcumin and 4-(cyclohexyl(phenyl)methoxy)-4-oxobutanoic acid (**S4**) in the presence of DMAP, EDC, and DCM produced difunctionalized (**7**, 16%), and monofunctionalized (**8**, 45%) succinate analogs (Figure 2).

On the other hand, when curcumin and 4-((9H-fluoren-9-yl)oxy)-4-oxobutanoic acid (**S5**) reacted in the presence of DMAP, EDC, and DCM, difunctionalized (**9**, 4%) and monofunctionalized (**10**, 20%) succinate analogs formed (Figure 2). However, when curcumin and 4-((2,3-dihydro-1H-inden-2-yl)oxy)-4-oxobutanoic acid (**S6**) reacted in the presence of DMAP, EDC, and Py, only the monofunctionalized (**11**, 16%) succinate analog was produced (Figure 2). The reaction between curcumin and 4-(((1*R*,2*S*,5*R*)-2-isopropyl-5-methylcyclohexyl)oxy)-4-oxobutanoic acid (**S7**) in the presence of DMAP, EDC, and DCM generated both difunctionalized (**12**, 9%), and monofunctionalized (**13**, 57%) succinate analogs (Figure 2). However, when curcumin was treated with 4-methoxy-4-oxobutanoic acid (**S8**) in the presence of DMAP, EDC, and DCM, only the monofunctionalized (**14**, 11%) succinate analog was produced (Figure 2). Finally, when curcumin reacted with 4-oxo-4-((1,2,3,4-tetrahydronaphthalen-1-yl)oxy)butanoic acid (**S9**) in the presence of DMAP, EDC, and DCM, only the monofunctionalized (**15**, 4%) succinate analog formed (Figure 2). The characterization of compounds **S1-S9** and **2-15** is available in the Appendix A.

### 2.2. Anti-Inflammatory Activity of the Curcumin Derivatives In Vitro

To evaluate the anti-inflammatory activity of curcumin and its derivatives (**2**–**15**), we measured the production of inflammatory mediators by murine macrophages stimulated with LPS in the presence or absence of the compounds. We previously reported a curcumin derivative **2,** which preserves the effect of curcumin on IL-6 production [16]. That compound and curcumin (**1**) were used as references when determining activity in this study. We first evaluated the effect of a single concentration of each compound on the production of IL-6 and TNF-α. Monofunctionalized compounds **4**, **6**, **10,** and **13**–**15** inhibited the production of IL-6 at 30 μM. Although the effect of compound **8** was not statistically significant, we observed a conserved trend throughout the experiments. Difunctionalized compounds **3**, **7**, **9**, **11**, and **12** did not show this activity (Figure 3B). In our experimental conditions, none of the compounds influenced TNF-α production (Figure 3A). Compounds **2**–**13** and **15** were not cytotoxic at this concentration (Figure 3C). However, under our experimental conditions, **1** and **14** affected cell viability. All monofunctionalized compounds were selected for further experiments.

We then evaluated the effects of the monofunctionalized compounds on IL-6 production at different concentrations. All compounds, including compounds **1** and **2**, inhibited the production of IL-6 in a dose-dependent manner (Figure 4). Greater inhibitory effects were observed at 10 and 30 µM, although compounds **1** and **2** still showed stronger activity. In general, at lower concentrations (1 and 3 µM), the effects of the compounds were similar to that of compound **1**. Compound **2** exhibited a statistically significant inhibitory effect even at the lowest concentration (1 µM), which was better than that of the rest of the compounds, including compound **1** (Figure 4). The effect of the compounds was not due to cellular death, as none of the compounds were cytotoxic at any of the evaluated concentrations.

In addition to IL-6, we evaluated whether the compounds could reduce the production of prostaglandin E_2_ (PGE_2_). PGE_2_ is secreted by macrophages that have been exposed to inflammatory stimuli, and its synthesis is influenced by the enzyme COX-2 [29]. It has been shown that curcumin can suppress COX-2 expression and PGE_2_ production in models of inflammation [30]. We decided to examine whether these curcumin derivatives, including compound **2**, preserved the effect on PGE_2_ production. We stimulated macrophages with LPS in the presence or absence of compounds and determined the levels of PGE_2_ six hours after stimulus. All monofunctionalized compounds inhibited the production of PGE_2_ in a dose-dependent manner, and the effect was statistically significant at concentrations of 10 and 30 μM (Figure 5). Among these, only compound **15** significantly reduced PGE_2_ production at a concentration of 3 μM. Compound **2** also showed an inhibitory effect on PGE_2_ production at all concentrations tested. At a concentration of 1 μM, only compound **2** showed an effect (Figure 5).

The IC_50_ values representing the effects of all of the compounds are listed in Table 1 and range from 1.94 ± 0.66 μM to 10.6 ± 0.33 μM for the inhibition of IL-6 production and from 0.51 ± 0.08 μM to 5.93 ± 2.29 μM for the effect on PGE_2_ production.

## 3. Discussion

Curcumin derivatives were prepared by first considering forming succinate and then esterifying curcumin.

When macrophages are activated, they release a wide variety of mediators, including proinflammatory cytokines such as TNF-α, IL-6, and prostaglandins (such as PGE_2_). These mediators are usually used as markers of an inflammatory response. We determined the anti-inflammatory effects of curcumin and its derivatives in macrophages stimulated by LPS, an inducer of inflammation. An anti-inflammatory effect was observed after treatment with the monofunctionalized curcumin derivatives with statistically significant dose-dependent inhibition of IL-6 production.

A structure–activity relationship (SAR) evaluation indicated that the difunctionalized compounds lost the anti-inflammatory activity of curcumin, while this property was maintained with the monofunctionalized compounds. Compounds **6**, **8**, and **10** showed the best activity among the monofunctionalized, with IC_50_ values of 4.21 ± 0.73, 1.94 ± 0.66, and 3.60 ± 0.21, respectively. Curcumin has activity comparable to those previously mentioned. Compound **2** exhibited the best activity with an IC_50_ of 3.59 ± 0.27. The compounds that were less active were **4**, **5**, **13,** and **15**. The structural difference between compound **2** and this new set of curcumin derivatives is the succinate linker between the curcumin structure and the ligand (benzyl alcohol). The addition of this linker to the new derivatives decreased the anti-inflammatory activity compared to compound **2**. These results indicate that the overall length of the derivative structure might influence activity [16].

The ligands of compounds **2**, **6**, **8**, and **10** all contain aromatic rings in the benzylic position. Among the compounds with two aromatic rings attaching to the benzylic position, **6** and **10** showed the greater effect on IL-6 production. However, when a single aromatic ring was attached to this position (**5**, **15**), the activity decreased [16].

A common feature of compounds **4**, **13**, and **15** is an aliphatic six-membered ring, and these compounds showed a lower activity than curcumin, indicating that the six-membered ring attached directly to the keto group of the ligand induces a loss in activity. However, when a six-membered aromatic ring is attached at a benzylic position (**8**), the activity of the compound increases. These results suggest that there are three key points to increasing anti-inflammatory activity: (i) the phenolic ring of the unesterified curcumin derivatives can form hydrogen bonds with another molecule, affecting the activity of these compounds; (ii) a π-π interaction with the curcumin ligand is possible since compounds **2**, **6**, **8**, and **10** showed high anti-inflammatory activity against IL-6, while compounds **4**, **13**, and **15** had low activity. (iii) The length of the curcumin ligand can influence its activity, as observed with compound **2** compared to compounds **6**, **8**, and **10**, in which the succinate linker in the latter compounds had an increased length and decreased activity compared to compound **2**.

Several studies have shown that nonsteroidal anti-inflammatory drugs (NSAIDs) inhibit LPS-stimulated PGE_2_ production [31]. Because curcumin is considered an NSAID, we evaluated the effect of this polyphenol and its derivatives on PGE_2_ production. A significant reduction in PGE_2_ secretion was observed, suggesting that these curcumin derivatives have a broad effect on the production of proinflammatory mediators associated with the LPS response.

Compound **2** was the most active compound, with a PGE_2_ IC_50_ value of 0.51 μM ± 0.08. Among the monofunctionalized, compounds **5**, **6**, and **15** showed activity, while compounds **4**, **8**, **10**, and **13** exhibited the lowest activity.

For PGE_2_, structure–activity relationships similar to like those found with IL-6 were observed, as a free hydroxyl group on the aromatic ring of curcumin enhanced activity. Additionally, the length of the molecule remained an influential factor when the aromatic rings in the benzylic position were present. The curcumin derivatives inhibited the production of IL-6 and PGE_2_ without affecting TNF-α secretion. NSAIDs suppress the synthesis of PGE_2_ by a mechanism that involves regulation of COX-2 activity [31]. Since curcumin derivatives presented herein affect the production of PGE_2_, further studies are necessary to elucidate a mechanism engaged in this effect. 

These results lead us to infer that the protection of both sides of the aromatic ring results in the inactivity, seemingly because of degradation. However, protection of one of the rings maintains the activity, and, depending on the type of the substituent, it can enhance it to the point of exceeding the activity of the unprotected molecule, contradicting the above conclusion. Thus, it remains unclear whether it suffers degradation in the unprotected ring. Given the uncertainty, we plan evaluate the mechanism of action of the monofunctionalized compounds in the near future and continue our search for protective groups (succinyl or ether) that are more favorable for prevention of the degradation of the curcumin derivative. At the moment, it can be seen that curcumin has in vitro activity when it is protected on only one side, and that the ether group is the most active.

## 4. Materials and Methods

### 4.1. Synthesis

Chemical reagents were used as commercially available (Tedia, Applichem, Sigma Aldrich, USA). All reactions were conducted in a borosilicate glass tube (20 mL or 16 mL) fitted with screw-cap and magnetic stirring under the atmosphere of argon. The reaction mixture was evaluated in an Agilent 1260 Infinity II HPLC system equipped with a quaternary pump, an Agilent diode array detector 1260 Series and normal phase silica gel column (Phenomenex^®^ Luna Silica (2), 250 mm × 10 mm, 5 μm) with gradient system *n*-hexane to ethyl acetate in 20 min at 2 mL/min (Agilent Technologies, Santa Clara, CA, USA). The reaction mixture was purified using a Buchi C-815 Flash HPLC system equipped with a binary pump, a UV scan withan Evaporative Light Scattering (ELSD) detector, a closed fraction collector bay, and normal phase silica gel cartridge (Buchi^®^ FlashPure EcoFlex Silica 12 g, 40–63 μm) in an isocratic system *n*-hexane/ethyl acetate 7:3 (BÜCHI Labortechnik AG, Meierseggstrasse 40 postfach, Switzerland). NMR characterization was performed on ^1^H. ^13^C NMR spectra were recorded at 500 (^1^H), 125 MHz (^13^C) on a Jeol JNM-ECZ500R/S1 500 MHz spectrometer in CDCl_3_ and DMSO-d_6_ (JEOL Ltd. 3-1-2 Musashino, Akishima, Tokyo 196-8558, Japan). Chemical shifts (δ) are reported in parts per million (ppm) from the residual solvent peak and coupling constant (*J*) in Hz. Proton multiplicity is reported in: singlet (s), doublet (d), triplet (t), quartet (quart.), quintet (quint.), septet (sept.), multiplet (m), broad (br). Infrared measurements were carried out on a Bruker Platinum ATR Alpha instrument (Bruker, Billerica, MA, USA). The MS analyses were carried out on a Waters Xevo TQD spectrometer with Electrospray Ionization (ESI) as an ion source (Waters Corporation, Milford, MA, USA). Melting point determinations were measured by triplicate on an Automatic Melting Point apparatus Stuart SMP50 (Cole-Palmer, Staffordshire, UK). The detailed synthesis procedure and spectral characterization are described below.

#### 4.1.1. General Procedure 1 (GP1) for the Synthesis of Alkyl Succinate Monoesters (**S1**–**S9**)

An oven dried vial (20 mL) fitted with screw-cap with magnetic stirrer was flushed with argon and charged with alcohol (854 mg, 14.7 mmol), dichloromethane (5 mL), and N,N-diisopropylethylamine (1.3 mL, 7.35 mmol, 0.5 equiv.) at room temperature (rt). After 2 h, succinic anhydride (735 mg, 7.35 mmol, 0.5 equiv.), and 4-dimethylaminopyridine (448 mg, 3.67 mmol, 0.25 equiv.) were added, and the reaction stirred at rt. After 48 h, the reaction mixture was diluted with brine/1M HCl (3:1, 10 mL). The aqueous layer was extracted with dichloromethane (3 × 20 mL), and the combined organic phases dried over anhydrous sodium sulfate (Na_2_SO_4_) and concentrated under reduced pressure.

##### Synthesis of 4-((Adamantan-2-yl)oxy)-4-oxobutanoic Acid (**S1**)

According to GP1, adamantan-2-ol (500.0 mg, 3.28 mmol), and N,N-diisopropylethylamine (858 μL, 4.92 mmol, 1.5 equiv.) were stirred in pyridine (4 mL) at rt. After 2 h, succinic anhydride (492.9 mg, 4.92 mmol, 1.5 equiv.) and 4-dimethylaminopyridine (601.8 mg, 4.92 mmol, 1.5 equiv.) were added, and the reaction mixture stirred for 48 h at rt to obtain the monoester S1 (713.3 mg, 86%). ^1^H NMR (500 MHz, DMSO-D_6_): δ 4.76 (t, J = 3.2 Hz, 1H), 2.49–2.44 (m, 5H), 1.95–1.84 (m, 4H), 1.79–1.63 (m, 8H), 1.47 (d, J = 11.5 Hz, 2H) ppm. ^13^C NMR (125 MHz, DMSO-D6): 26.4, 26.6, 28.9, 29.3, 31.2, 35.7, 36.8, 76.2, 171.3, 173.5. IR: 2901.31, 1702.61, 1699.69, 1322.74, 1162.63. mp: 87.8 ± 1.5 °C. MS (m/z) calcd for C_14_H_20_NaO_4_: 275.30; found: 274.99 [M+Na^+^].

##### Synthesis of 4-((9,10-Dioxo-9,10-dihydroanthracen-2-yl)methoxy)-4-oxobutanoic Acid (**S2**)

According to GP1, 2-(hydroxymethyl)anthracene-9,10-dione (200.9 mg, 0.84 mmol), and N,N-diisopropylethylamine (500 μL, 2.87 mmol, 3.4 equiv.) were stirred in CH_2_Cl_2_ (5 mL) at rt. After 2 h, succinic anhydride (251.9 mg, 2.52 mmol, 3.0 equiv.) and 4-dimethylaminopyridine (307.6 mg, 2.52 mmol, 3.0 equiv.) were added, and the reaction mixture stirred for 48 h at RT to yield monoester S2 (255.2 mg, 89%). ^1^H NMR (500 MHz, DMSO-D_6_): δ 8.14–8.05 (m, 4H), 7.87–7.82 (m, 2H), 7.79 (d, J = 9.7 Hz, 1H), 5.23 (s, 2H), 2.61–2.56 (m, 2H), 2.49–2.44 (m, 2H) ppm. ^13^C NMR (125 MHz): 28.72, 28.74, 64.5, 125.4, 126.7, 126.8, 127.1, 132.5, 133.01, 133.04, 133.0, 133.1, 134.6, 134.7, 143.2, 172.1, 173.5, 182.2, 182.3 ppm. IR: 2932.30, 1684.14, 1671.36, 1652.92, 1591.01, 1175.53, 705.69. mp: 152.1 ± 1.1 °C.

##### Synthesis of 4-(Benzhydryloxy)-4-oxobutanoic Acid (**S3**)

According to GP1, diphenylmethanol (1380.6 mg, 7.49 mmol), and N,N-diisopropylethylamine (870 μL, 5.00 mmol, 0.67 equiv.) were stirred in CH_2_Cl_2_ (4 mL) at rt. After 2 h, succinic anhydride (500 mg, 5.00 mmol, 0.67 equiv.) and 4-dimethylaminopyridine (610.4 mg, 5.00 mmol, 0.67 equiv.) were added, and the reaction mixture stirred for 48 h at rt to yield monoester S3 (1865.3 mg, 88%). ^1^H NMR (500 MHz, DMSO-D_6_): δ 7.35–7.28 (m, 10H), 6.75 (s, 1H), 2.64–2.60 (m, 2H), 2.50–2.47 (m, 2H). ^13^C NMR (125 MHz, DMSO-D_6_): δ 173.39, 171.19, 140.57, 128.48, 127.70, 126.48, 76.45, 29.02, 28.67 ppm. IR: 3059.77, 3028.62, 1733.35, 1240.98, 1154.35, 694.86. MS (m/z) calcd for C_17_H_16_NaO_4_: 307.3; found: 306.9 [M+Na^+^].

##### Synthesis of 4-(Cyclohexyl(phenyl)methoxy)-4-oxobutanoic Acid (**S4**)

According to GP1, cyclohexyl(phenyl)methanol (50.0 mg, 0.26 mmol), and N,N-diisopropylethylamine (100 μL, 0.57 mmol, 2.1 equiv.) were stirred in CH_2_Cl_2_ (2 mL) at rt. After 2 h, succinic anhydride (80.0 mg, 0.80 mmol, 3.0 equiv.) and 4-dimethylaminopyridine (100 mg, 0.82 mmol, 3.0 equiv.) were added, and the reaction mixture stirred for 48 h at rt to yield monoester S4 (74.8 mg, 98%). ^1^H NMR (500 MHz, DMSO-D_6_): δ 7.35–7.24 (m, 5H), 5.45 (d, J = 7.1 Hz, 1H), 2.58–2.52 (m, 2H), 2.49–2.45 (m, 2H), 1.75–1.55 (m, 5H), 1.31–0.93 (m, 6H) ppm.^13^C NMR (125 MHz, DMSO-D_6_): 25.3, 25.4, 25.8, 28.0, 28.6, 28.7, 28.9, 29.0, 42.5, 79.3, 126.6, 127.5, 128.1, 139.5, 171.3, 173.3, 173.6 ppm. IR: 2916.61, 2846.67, 1721.37, 1225.98, 1164.43, 695.52 cm^−1^. mp: 156.1 ± 3.5 °C. MS (m/z) calcd for C_17_H_22_NaO_4_: 313.3; found: 313.0 [M+Na^+^].

##### Synthesis of 4-((9H-Fluoren-9-yl)oxy)-4-oxobutanoic Acid (**S5**)

According to GP1, 9H-fluoren-9-ol (500 mg, 2.74 mmol), and *N,N*-diisopropylethylamine (717 μL, 4.12 mmol, 1.5 equiv.) were stirred in pyridine (4 mL) at rt. After 2 h, succinic anhydride (411.9 mg, 4.12 mmol, 1.5 equiv.) and 4-dimethylaminopyridine (502.8 mg, 4.12 mmol, 1.5 equiv.) were added, and the reaction mixture stirred for 48 h at rt to yield monoester S5 (600 mg, 77%). ^1^H NMR (500 MHz, DMSO-*D*_6_): δ 7.84 (d, *J* = 7.5 Hz, 2H), 7.52 (dd, *J* = 7.5, 1.0 Hz, 2H), 7.50–7.41 (m, 2H), 7.33 (td, *J* = 7.4, 1.1 Hz, 2H), 6.75 (s, 1H), 2.66–2.62 (m, 2H), 2.58–2.55 (m, 2H) ppm. ^13^C NMR (125 MHz, DMSO-*D*_6_): δ 173.6, 173.4, 141.8, 140.4, 129.6, 127.9, 125.7, 120.4, 74.5, 28.9. mp: 130.5 ± 0.1 °C. MS (m/z) calcd for C_17_H_14_NaO_4_: 305.2; found: 304.9 [M+Na^+^].

##### Synthesis of 4-((2,3-Dihydro-1H-inden-2-yl)oxy)-4-oxobutanoic Acid (**S6**)

According to GP1, 2,3-dihydro-1H-inden-2-ol (500 mg, 3.73 mmol), and *N,N*-diisopropylethylamine (973 μL, 5.59 mmol, 1.5 equiv.) were stirred in CH_2_Cl_2_ (4 mL) at rt. After 2 h, succinic anhydride (559.3 mg, 5.59 mmol, 1.5 equiv.) and 4-dimethylaminopyridine (682.8 mg, 5.59 mmol, 1.5 equiv.) were added, and the reaction mixture stirred for 48 h at RT to yield monoester S6 (817.8 mg, 94%). ^1^H NMR (500 MHz, DMSO): δ 12.19 (s, 1H), 7.27–7.14 (m, 4H), 5.46–5.40 (m, 1H), 3.27 (dd, *J* = 17.1, 6.4 Hz, 2H), 2.89 (d, *J* = 17.0 Hz, 2H), 2.44 (s, 4H) ppm. ^13^C NMR (125 MHz, DMSO): δ 173.4, 172.1, 140.4, 126.6, 124.5, 75.0, 28.8, 28.6 ppm. IR: 2943.05, 2910.11, 1716.85, 1652.88, 1398.03, 1165.52, 941.71, 748.10. mp: 112.8 ± 0.9 °C. MS (m/z) calcd for C_13_H_14_NaO_4_: 257.2; found: 256.9 [M+Na^+^].

##### Synthesis of 4-(((1R,2S,5R)-2-Isopropyl-5-methylcyclohexyl)oxy)-4-oxobutanoic Acid (**S7**)

According to GP1, (1R,2S,5R)-(-)Menthol (500.0 mg, 3.20 mmol), and *N,N*-diisopropylethylamine (836 μL, 4.80 mmol, 1.5 equiv.) were stirred in CH_2_Cl_2_ (4 mL) at rt. After 2 h, succinic anhydride (480.2 mg, 4.80 mmol, 1.5 equiv.) and 4-dimethylaminopyridine (586.3 mg, 4.80 mmol, 1.5 equiv.) were added, and the reaction mixture stirred for 48 h at rt to yield monoester S7 (774.1 mg, 94%). ^1^H NMR (500 MHz, DMSO): δ 4.63–4.54 (m, 1H), 2.47 (s, 4H), 1.90–1.79 (m, 2H), 1.67–1.58 (m, 2H), 1.50–1.28 (m, 2H), 1.08–0.90 (m, 2H), 0.90–0.83 (m, 7H), 0.71 (d, *J* = 6.9 Hz, 3H). ^13^C NMR (125 MHz, DMSO): δ 73.2, 46.4, 40.5, 33.7, 30.8, 29.0, 28.7, 25.7, 23.0, 21.9, 20.5, 16.3. IR: 2949.05, 2868.19, 1706.67, 1699.11, 1652.74, 1387.20, 1169.71, 952.45. mp: not determined. MS (m/z) calcd for C_14_H_24_NaO_4_: 279.33; found: 279.03 [M+Na^+^].

##### Synthesis of 4-Methoxy-4-oxobutanoic Acid (**S8**)

According to GP1, methanol (616.8 μL, 15.6 mmol), and *N,N*-diisopropylethylamine (2249 μL, 12.9 mmol, 0.8 equiv.) were stirred in CH_2_Cl_2_ (4 mL) at rt. After 2 h, succinic anhydride (1292.2 mg, 12.9 mmol, 0.8 equiv.) and 4-dimethylaminopyridine (1577.6 mg, 12.9 mmol, 0.8 equiv.) were added, and the reaction mixture stirred for 48 h at rt to yield monoester S8 (1381.0 mg, 67%). ^1^H NMR (500 MHz, DMSO-*D*_6_): δ 12.16 (s, 1H), 3.55 (s, 3H), 2.48–2.41 (m, 4H) ppm. ^13^C NMR (125 MHz, DMSO-*D*_6_): δ 173.4, 172.6, 51.4, 28.6, 28.5 ppm. IR: 2932.26, 1732.32, 1653.16, 1436.81, 1170.72, 942.87. mp: 56.8 ± 0.6 °C. MS (m/z) calcd for C_5_H_8_NaO_4_: 155.11; found: 154.93 [M+Na^+^].

##### Synthesis of 4-oxo-4-((1,2,3,4-Tetrahydronaphthalen-1-yl)oxy)butanoic Acid (**S9**)

According to GP1, 1,2,3,4-tetrahydronaphthalen-1-ol (500 mg, 3.37 mmol), and *N,N*-diisopropylethylamine (881 μL, 5.06 mmol, 1.50 equiv.) were stirred in CH_2_Cl_2_ (4 mL) at rt. After 2 h, succinic anhydride (506.4 mg, 5.06 mmol, 1.5 equiv.) and 4-dimethylaminopyridine (618.3 mg, 5.06 mmol, 1.50 equiv.) were added, and the reaction mixture stirred for 48 h at rt to yield monoester S11 (700.6 mg, 84%). ^1^H NMR (500 MHz, DMSO): δ12.23 (s, 1H), 7.26–7.09 (m, 4H), 5.90–5.84 (m, 1H), 2.84–2.61 (m, 2H), 2.55–2.43 (m, 5H), 1.97–1.71 (m, 4H).^13^C NMR (125 MHz, DMSO): δ 173.4, 171.8, 137.6, 134.4, 129.0, 128.8, 127.9, 125.9, 69.4, 29.1, 28.8, 28.6, 28.3, 18.5. mp: 91.8 ± 0.2 °C. MS (m/z) calcd for C_14_H_16_NaO_4_: 271.27; found: 270.99 [M+Na^+^].

#### 4.1.2. General Procedure (GP2) for the Synthesis of Dialkylcurcumin and Monoalkylcurcumin (**3**–**15**)

A borosilicate glass tube (16 mL) fitted with screw-caps equipped with magnetic stirrer was flushed with argon and charged with alkyl succinate (41.4 mg, 0.12 mmol, 0.9 equiv.), 4-dimethylaminopyridine (15.1 mg, 0.12 mmol, 1.0 equiv.), curcumin (50.1 mg, 0.13 mmol), and dichloromethane (4 mL), which were combined and stirred at 0 °C for 10 min. After, N-(3-Dimethylaminopropyl)-N′-ethylcarbodiimide hydrochloride (24.9 mg, 0.13 mmol, 1.0 equiv.) was added and stirred at rt for 24 h. The reaction was extracted with with EtOAc and water (3 × 20 mL). The organic phases were concentrated under reduced pressure, and the remaining material was purified by HPLC to obtain monoester (11.3 mg, 13%).

##### Synthesis of Di((1r,3r,5r,7r)-adamantan-2-yl) O,O′-(((1E,3Z,6E)-3-hydroxy-5-oxohepta-1,3,6-triene-1,7-diyl)bis(2-methoxy-4,1-phenylene)) Disuccinate (**3**) and (1r,3r,5r,7r)-Adamantan-2-yl (4-((1E,4Z,6E)-5-hydroxy-7-(4-hydroxy-3-methoxyphenyl)-3-oxohepta-1,4,6-trien-1-yl)-2-methoxyphenyl) Succinate (**4**)

According to GP2, succinate S1 (31.3 mg, 0.12 mmol, 1.0 equiv.), 4-dimethylaminopyridine (16.1 mg, 0.13 mmol, 1.0 equiv.), curcumin (50.5 mg, 0.13 mmol), and dichloromethane (4 mL) were combined and stirred at 0 °C for 10 min. After, N-(3-Dimethylaminopropyl)-N′-ethylcarbodiimide hydrochloride (26.1 mg, 0.14 mmol, 1.0 equiv.) was added and stirred at rt for 24 h. The crude product was purified by HPLC to obtain diester 3 (21.0 mg, 37%) and monoester 4 (25.5 mg, 62%). Compound 3: ^1^H NMR (500 MHz, CDCl_3_): δ 7.62 (d, *J* = 15.9 Hz, 2H), 7.17–7.06 (m, 6H), 6.57 (d, *J* = 15.8 Hz, 2H), 5.89–5.83 (m, 1H), 4.97 (s, 2H), 3.87 (s, 6H), 2.96 (t, *J* = 7.4 Hz, 4H), 2.79 (t, *J* = 6.9 Hz, 4H), 2.01 (s, 8H), 1.90–1.66 (m, 20H). ^13^C NMR (125 MHz, CDCl_3_): δ 183.2, 171.5, 170.5, 151.5, 141.3, 140.1, 134.1, 124.4, 123.4, 121.2, 111.5, 102.0, 77.7, 56.1, 37.5, 36.4, 31.9, 31.9, 29.8, 29.2, 27.3, 27.1.ppm. IR: 2909.0, 2854.6, 1765.5, 1729.4, 1630.3, 1508.2, 1416.5, 1256.1, 1129.7 cm^−1^. MS (m/z) calcd for C_49_H_56_NaO_12_: 859.37; found: 859.6 [M+Na^+^]. Compound 4:^1^H NMR (500 MHz, CDCl_3_): δ 7.60 (dd, *J* = 15.8, 5.7 Hz, 2H), 7.19–7.00 (m, 6H), 6.94 (d, *J* = 8.2 Hz, 1H), 6.52 (dd, *J* = 27.3, 15.8 Hz, 2H), 5.83 (s, 1H), 5.01–4.93 (m, 1H), 3.95 (s, 3H), 3.87 (s, 3H), 2.96 (t, *J* = 7.0 Hz, 2H), 2.79 (t, *J* = 7.0 Hz, 2H), 2.00 (s, 4H), 1.87–1.71 (m, 10H). ^13^C NMR (125 MHz, CDCl_3_): δ 184.6, 181.9, 171.5, 170.6, 151.4, 148.1, 146.9, 141.3, 141.2, 139.6, 134.2, 127.7, 124.3, 123.4, 123.2, 121.9, 121.1, 115.0, 111.5, 109.7, 101.7, 77.7, 77.4, 56.1, 56.0, 37.5, 36.4, 31.9, 31.9, 29.8, 29.2, 27.3, 27.1 ppm. IR: 3403.0, 2908.8, 2854.7, 1726.1, 1626.5, 1587.5, 1509.6, 1267.0, 1129.0 cm^−1^. MS (m/z) calcd for C_35_H_38_NaO_9_: 625.24; found: 625.4 [M+Na^+^].

##### Synthesis of (9,10-Dioxo-9,10-dihydroanthracen-2-yl)methyl (4-((1E,4Z,6E)-5-hydroxy-7-(4-hydroxy-3-methoxyphenyl)-3-oxohepta-1,4,6-trien-1-yl)-2-methoxyphenyl) Succinate (**5**)

According to GP2, succinate S2 (41.4 mg, 0.12 mmol, 0.9 equiv), 4-dimethylaminopyridine (15.1 mg, 0.12 mmol, 1.0 equiv.), curcumin (50.1 mg, 0.13 mmol), and dichloromethane (4 mL) were combined and stirred at 0 °C for 10 min. After, N-(3-Dimethylaminopropyl)-N′-ethylcarbodiimide hydrochloride (24.9 mg, 0.13 mmol, 1.0 equiv.) was added and stirred at rt for 24 h. The crude product was purified by HPLC to obtain monoester 5 (11.3 mg, 13%). Compound 5: ^1^H NMR (500 MHz, CDCl_3_): δ 8.31–8.26 (m, 4H), 7.84–7.76 (m, 4H), 7.56 (dd, *J* = 50.9, 15.8 Hz, 2H), 7.14–7.04 (m, 4H), 6.96 (dd, *J* = 23.4, 8.2 Hz, 2H), 6.49 (dd, *J* = 15.8, 5.2 Hz, 2H), 5.81 (s, 1H), 5.32 (s, 2H), 3.95 (s, 3H), 3.84 (s, 3H), 3.00–2.96 (m, 2H), 2.89–2.85 (m, 2H) ppm.—^13^C NMR (125 MHz, CDCl_3_): 29.1, 29.3, 56.0, 56.1, 65.6, 101.7, 109.7, 111.5, 115.0, 126.4, 127.40, 127.43, 127.9, 133.3, 134.4, 139.4, 141.0, 141.3, 142.5, 146.9, 148.1, 151.3, 170.2, 171.9, 181.8, 182.9, 183.0, 184.7 ppm. IR: 3046.5, 2925.0, 2849.9, 1757.0, 1731.0, 1671.7, 1588.7, 1511.3, 1444.4, 1294.6, 1203.3, 1136.8 cm^−1^. MS (m/z) calcd for C_40_H_32_O_11_: 688.6; found: 689.4 [M+H^+^].

##### Synthesis of Benzhydryl (4-((1E,4Z,6E)-5-hydroxy-7-(4-hydroxy-3-methoxyphenyl)-3-oxohepta-1,4,6-trien-1-yl)-2-methoxyphenyl) Succinate (**6**)

According to GP2, succinate **S3** (135.88 mg, 0.48 mmol, 1.2 equiv), 4-dimethylaminopyridine (65.4 mg, 0.54 mmol, 1.3 equiv.), curcumin (152.4 mg, 0.41 mmol), and dichloromethane (4 mL) were combined and stirred at 0 °C for 10 min. After, N-(3-Dimethylaminopropyl)-N′-ethylcarbodiimide hydrochloride (84.87 mg, 0.44 mmol, 1.1 equiv.) was added and stirred at rt for 24 h. The crude product was purified by column chromatography on silica gel to obtain monoester 6 (37.7 mg, 14%). Compound 6: ^1^H NMR (500 MHz, CDCl_3_): δ 7.59 (dd, *J* = 15.8, 9.1 Hz, 2H), 7.35–7.30 (m, 10H), 7.13–7.03 (m, 4H), 6.93 (dd, *J* = 8.2, 4.6 Hz, 2H), 6.91 (s, 1H), 6.51 (dd, *J* = 24.4, 15.8 Hz, 2H), 5.82 (s, 1H), 3.94 (s, 3H), 3.79 (s, 3H), 2.97–2.92 (m, 2H), 2.90–2.86 (m, 2H) ppm. ^13^C NMR (125 MHz, CDCl_3_): δ 184.7, 181.9, 171.2, 170.3, 151.4, 148.1, 146.9, 141.3, 141.1, 140.1, 139.6, 134.2, 128.7, 128.1, 127.2, 124.3, 123.4, 123.2, 121.9, 121.1, 115.0, 111.4, 109.7, 101.7, 77.6, 77.4, 56.1, 56.0, 29.6, 29.1 ppm. IR: 3511.7, 2922.4, 2850.8, 1740.0, 1624.7, 1586.0, 1510.1, 1302.8, 1261.5, 1121.9 cm^−1^_._ MS (m/z) calcd for C_38_H_34_NaO_9_: 657.21; found: 657.4 [M+Na^+^].

##### Synthesis of Bis(cyclohexyl(phenyl)methyl) O,O′-(((1E,3Z,6E)-3-hydroxy-5-oxohepta-1,3,6-triene-1,7-diyl)bis(2-methoxy-4,1-phenylene)) Disuccinate (**7**) and Cyclohexyl(phenyl)methyl (4-((1E,4Z,6E)-5-hydroxy-7-(4-hydroxy-3-methoxyphenyl)-3-oxohepta-1,4,6-trien-1-yl)-2-methoxyphenyl) Succinate (**8**)

According to GP2, succinate S4 (114.8 mg, 0.40 mmol, 1.0 equiv), 4-dimethylaminopyridine (89.7 mg, 0.73 mmol, 1.8 equiv.), curcumin (150.3 mg, 0.40 mmol), and dichloromethane (4 mL) were combined and stirred at 0 °C for 10 min. After, N-(3-Dimethylaminopropyl)-N′-ethylcarbodiimide hydrochloride (92.8 mg, 0.48 mmol, 1.2 equiv.) was added and stirred at rt for 24 h. The crude product was purified by column chromatography on silica gel to obtain diester 7 (29.6 mg, 16%) and monoester 8 (59.2 mg, 45%). Compound 7: ^1^H NMR (500 MHz, CDCl_3_): δ 7.61 (d, J = 15.8 Hz, 2H), 7.33–7.27 (m, 10H), 7.17–7.09 (m, 4H), 6.99 (d, J = 8.1 Hz, 2H), 6.56 (d, J = 15.8 Hz, 2H), 5.52 (d, J = 8.0 Hz, 2H), 3.83 (s, 6H), 3.00–2.66 (m, 10H), 1.88–1.70 (m, 8H), 1.24–0.90 (m, 12H) ppm. ^13^C NMR (125 MHz, CDCl_3_): δ 183.2, 171.4, 170.3, 151.4, 141.3, 140.1, 139.6, 134.1, 128.3, 128.3, 127.9, 127.2, 127.2, 124.4, 123.4, 121.2, 111.5, 102.0, 81.0, 56.0, 43.1, 29.5, 29.1, 29.1, 26.3, 26.0, 25.9 ppm. IR: 2927.4, 2852.0, 1763.4, 1732.1, 1629.0, 1506.8, 1415.7, 1253.7, 1122.0 cm^−1^. MS (m/z) calcd for C_55_H_60_NaO_12_: 935.40; found: 935.8 [M+Na^+^]. Compound 8: ^1^H NMR (500 MHz, CDCl_3_): δ 7.59 (dd, J = 15.8, 12.5 Hz, 2H), 7.35–7.26 (m, 5H), 7.14–7.02 (m, 4H), 6.95 (dd, J = 26.7, 8.1 Hz, 2H), 6.51 (dd, J = 22.6, 15.8 Hz, 2H), 5.82 (s, 1H), 5.52 (d, J = 8.0 Hz, 1H), 3.92 (d, J = 0.9 Hz, 3H), 3.82 (s, 3H), 3.01–2.63 (m, 5H), 1.93–1.66 (m, 4H), 1.22–0.86 (m, 6H) ppm. ^13^C NMR (125 MHz, CDCl_3_): δ 184.6, 181.9, 171.4, 170.4, 151.4, 148.1, 146.9, 141.3, 141.1, 139.6, 139.5, 134.2, 128.3, 127.9, 127.2, 115.0, 111.4, 109.7, 101.7, 81.0, 56.0, 56.0, 43.0, 29.5, 29.1, 29.1, 29.0, 26.3, 25.9, 25.9 ppm. IR: 3430.5, 2930.5, 2851.9, 1763.9, 1733.4, 1627.0, 1587.9, 1510.0, 1267.5, 1130.9 cm^−1^. MS (m/z) calcd for C_38_H_40_NaO_9_: 663.26; found: 663.5 [M+Na^+^].

##### Synthesis of Di(9H-fluoren-9-yl) O,O′-(((1E,3Z,6E)-3-hydroxy-5-oxohepta-1,3,6-triene-1,7-diyl)bis(2-methoxy-4,1-phenylene)) Disuccinate (**9**) and 9H-Fluoren-9-yl (4-((1E,4Z,6E)-5-hydroxy-7-(4-hydroxy-3-methoxyphenyl)-3-oxohepta-1,4,6-trien-1-yl)-2-methoxyphenyl) Succinate (**10**)

According to GP2, succinate S5 (34.5 mg, 0.12 mmol, 0.9 equiv), 4-dimethylaminopyridine (15.7 mg, 0.13 mmol, 1.0 equiv.), curcumin (50.0 mg, 0.13 mmol), and dichloromethane (4 mL) were combined and stirred at 0 °C for 10 min. After, N-(3-Dimethylaminopropyl)-N′-ethylcarbodiimide hydrochloride (24.2 mg, 0.13 mmol, 1.0 equiv.) was added and stirred at rt for 24 h. The crude product was purified by HPLC to obtain diester 9 (4.4 mg, 4%) and monoester 10 (22.3 mg, 26%). Compound 9: ^13^C NMR (125 MHz, CDCl_3_): δ 183.2, 172.9, 170.4, 151.4, 148.0, 146.9, 141.9, 141.1, 134.1, 129.7, 129.7, 128.0, 128.0, 126.1, 126.0, 120.2, 120.2, 114.9, 111.5, 109.7, 75.6, 56.1, 56.0, 52.1, 29.6, 29.5, 29.2, 29.1 ppm. IR: 2958.3, 2923.4, 2852.1, 1760.7, 1730.9, 1627.7, 1587.8, 1505.3, 1451.3, 1249.6, 1117.3 cm^−1^. MS (m/z) calcd for C_55_H_44_NaO_12_: 919.27; found: 919.30 [M+Na^+^]. Compound 10: ^13^C NMR (125 MHz, CDCl_3_): δ 184.7, 181.9, 172.9, 170.4, 151.4, 148.1, 147.9, 146.9, 146.9, 141.9, 141.1, 129.7, 128.0, 126.1, 123.0, 121.9, 121.8, 120.2, 115.0, 114.9, 109.7, 101.8, 75.6, 56.1, 56.0, 52.1, 29.6, 29.2 ppm. IR: 3426.6, 2959.8, 2926.3, 2850.6, 1760.2, 1730.6, 1625.1, 1585.5, 1507.1, 1450.4, 1250.8, 1117.8 cm^−1^. MS (m/z) calcd for C_38_H_32_NaO_9_: 655.19; found: 655.05 [M+Na^+^].

##### Synthesis of Bis(2,3-dihydro-1H-inden-2-yl) O,O′-(((1E,3Z,6E)-3-hydroxy-5-oxohepta-1,3,6-triene-1,7-diyl)bis(2-methoxy-4,1-phenylene)) Disuccinate (**11**)

According to GP2, succinate S6 (100.2 mg, 0.43 mmol, 3.1 equiv), 4-dimethylaminopyridine (23.9 mg, 0.20 mmol, 1.2 equiv.), curcumin (50.1 mg, 0.14 mmol), and pyridine (2.5 mL) were combined and stirred at 0 °C for 10 min. After, N-(3-Dimethylaminopropyl)-N′-ethylcarbodiimide hydrochloride (152.1 mg, 0.79 mmol, 5.8 equiv.) was added and stirred at rt for 24 h. The crude product was purified by HPLC to obtain diester 11 (17.6 mg, 16%). Compound 11: ^1^H NMR (500 MHz, CDCl_3_): δ 7.62 (d, *J* = 15.8 Hz, 2H), 7.24–7.19 (m, 8H), 7.16–7.10 (m, 4H), 7.01 (d, *J* = 8.2 Hz, 2H), 6.57 (d, *J* = 15.8 Hz, 2H), 5.87 (s, 1H), 5.61–5.57 (m, 2H), 3.85 (s, 6H), 3.33 (dd, *J* = 16.9, 6.4 Hz, 4H), 3.03 (dd, *J* = 17.0, 2.9 Hz, 4H), 2.92 (t, *J* = 6.8 Hz, 4H), 2.71 (t, *J* = 6.8 Hz, 4H). ^13^C NMR (125 MHz, CDCl_3_): δ183.2, 172.1, 170.5, 151.4, 141.2, 140.5, 140.1, 134.1, 126.9, 124.8, 124.8, 124.8, 124.3, 123.4, 121.2, 111.5, 102.0, 75.9, 56.0, 39.7, 29.5, 29.0 ppm. IR: 2922.7, 2851.5, 1760.9, 1727.1, 1505.8, 1415.1, 1252.5, 1117.6 cm^−1^. MS (m/z) calcd for C_47_H_44_NaO_12_: 823.27; found: 823.5 [M+Na^+^]. 

##### Synthesis of O,O′-(((1E,3Z,6E)-3-Hydroxy-5-oxohepta-1,3,6-triene-1,7-diyl)bis(2-methoxy-4,1-phenylene)) bis((1R,2S,5R)-2-isopropyl-5-methylcyclohexyl) Disuccinate (**12**) and 4-((1E,4Z,6E)-5-Hydroxy-7-(4-hydroxy-3-methoxyphenyl)-3-oxohepta-1,4,6-trien-1-yl)-2-methoxyphenyl((1R,2S,5R)-2-isopropyl-5-methylcyclohexyl) Succinate (**13**)

According to GP2, succinate S7 (33.1 mg, 0.13 mmol, 1.0 equiv), 4-dimethylaminopyridine (17.5 mg, 0.14 mmol, 1.1 equiv.), curcumin (50.3 mg, 0.13 mmol), and dichloromethane (4 mL) were combined and stirred at 0 °C for 10 min. After, N-(3-Dimethylaminopropyl)-N′-ethylcarbodiimide hydrochloride (23.7 mg, 0.12 mmol, 0.9 equiv.) was added and stirred at rt for 24 h. The crude product was purified by HPLC to obtain diester 12 (5.3 mg, 9%) and monoester 13 (23.6 mg, 57%). Compound 12: ^1^H NMR (500 MHz, CDCl_3_): δ 7.62 (d, *J* = 15.8 Hz, 2H), 7.17–7.06 (m, 6H), 6.57 (d, *J* = 15.8 Hz, 2H), 5.86 (s, 1H), 4.75–4.70 (m, 2H), 3.87 (s, 6H), 2.93 (t, *J* = 7.2 Hz, 4H), 2.75–2.72 (m, 4H), 1.99 (d, *J* = 11.5 Hz, 2H), 1.88–1.84 (m, 2H), 1.68 (dd, *J* = 8.9, 5.4 Hz, 4H), 1.46–1.35 (m, 4H), 1.03–0.96 (m, 4H), 0.89–0.87 (m, 12H), 0.74 (d, *J* = 6.9 Hz, 6H). ^13^C NMR (125 MHz, CDCl_3_): δ 183.2, 171.7, 170.5, 151.5, 141.3, 140.1, 134.1, 124.4, 123.4, 121.2, 111.6, 102.0, 74.9, 56.1, 47.1, 41.0, 34.3, 31.5, 29.6, 29.2, 26.4, 23.5, 22.2, 20.9, 16.4 ppm. IR: 2954.7, 2929.4, 2869.3, 1765.2, 1727.9, 1630.7, 1508.4, 1462.4, 1416.6, 1129.9 cm^−1^. MS (m/z) calcd for C_49_H_64_NaO_12_: 867.43; found: 867.46 [M+Na^+^]. Compound 13: ^1^H NMR (500 MHz, CDCl_3_): δ 7.60 (dd, *J* = 15.8, 7.0 Hz, 2H), 7.18–7.01 (m, 6H), 6.93 (d, *J* = 8.2 Hz, 1H), 6.51 (dd, *J* = 26.9, 15.7 Hz, 2H), 5.82 (s, 1H), 4.79–4.65 (m, 1H), 3.94 (s, 3H), 3.86 (s, 3H), 2.96–2.90 (m, 2H), 2.78–2.69 (m, 2H), 2.02–1.96 (m, 1H), 1.86 (qd, *J* = 7.0, 2.7 Hz, 1H), 1.70–1.64 (m, 2H), 1.49–1.34 (m, 2H), 1.29–1.21 (m, 1H), 1.08–0.95 (m, 2H), 0.88 (t, *J* = 6.9 Hz, 6H), 0.74 (d, *J* = 7.0 Hz, 3H) ppm. ^13^C NMR (125 MHz, CDCl_3_): δ 184.7, 181.9, 171.7, 170.5, 151.4, 148.1, 146.9, 141.3, 141.2, 139.5, 134.2, 127.6, 124.3, 123.4, 123.2, 121.8, 121.1, 115.0, 111.5, 109.7, 101.7, 74.9, 56.1, 56.0, 47.1, 41.0, 34.3, 31.5, 29.6, 29.2, 26.3, 23.5, 22.1, 20.9, 16.4 ppm. IR: 2954.5, 2931.6, 2869.1, 1764.7, 1726.6, 1627.4, 1510.3, 1417.7, 1267.7, 1131.6 cm^−1^. MS (m/z) calcd for C_35_H_42_NaO_9_: 629.27; found: 629.17 [M+Na^+^].

##### Synthesis of 4-((1E,4Z,6E)-5-Hydroxy-7-(4-hydroxy-3-methoxyphenyl)-3-oxohepta-1,4,6-trien-1-yl)-2-methoxyphenyl Methyl Succinate (**14**)

According to GP2, succinate S8 (60.4 mg, 0.46 mmol, 1.1 equiv), 4-dimethylaminopyridine (66.4 mg, 0.54 mmol, 1.3 equiv.), curcumin (151.6 mg, 0.41 mmol), and dichloromethane (4 mL) were combined and stirred at 0 °C for 10 min. After, N-(3-Dimethylaminopropyl)-N′-ethylcarbodiimide hydrochloride (87.9 mg, 0.46 mmol, 1.1 equiv.) was added and stirred at rt for 24 h. The crude product was purified by column chromatography on silica gel to obtain monoester 14 (22.3 mg, 11%). Compound 14: ^1^H NMR (500 MHz, CDCl_3_): δ 7.65–7.52 (m, 2H), 7.16–7.00 (m, 5H), 6.92 (dd, *J* = 8.1, 1.8 Hz, 1H), 6.56–6.42 (m, 2H), 5.82 (s, 1H), 3.93 (s, 3H), 3.86 (s, 3H), 3.72 (s, 3H), 2.94 (t, *J* = 6.7 Hz, 2H), 2.76 (t, *J* = 6.9 Hz, 2H) ppm. ^13^C NMR (125 MHz, CDCl_3_): δ 184.7, 181.8, 172.6, 170.5, 151.4, 148.3, 148.1, 147.0, 147.0, 141.3, 141.1, 140.7, 139.5, 134.2, 127.6, 127.5, 124.3, 123.4, 123.2, 123.0, 121.7, 121.7, 121.1, 115.0, 115.0, 111.5, 109.8, 109.7, 106.5, 101.7, 56.0, 52.1, 29.0 ppm. IR: 3405.4, 2950.9, 2845.6, 1760.1, 1735.3, 1625.9, 1586.8, 1509.7, 1417.8, 1126.9 cm^−1^. MS (m/z) calcd for C_35_H_34_NaO_9_: 621.21; found: 621.4 [M+Na^+^].

##### Synthesis of 4-((1E,4Z,6E)-5-Hydroxy-7-(4-hydroxy-3-methoxyphenyl)-3-oxohepta-1,4,6-trien-1-yl)-2-methoxyphenyl (1,2,3,4-tetrahydronaphthalen-1-yl) Succinate (**15**)

According to GP2, succinate S9 (95.91 mg, 0.39 mmol, 1.0 equiv), 4-dimethylaminopyridine (52.9 mg, 0.43 mmol, 1.1 equiv.), curcumin (150.47 mg, 0.41 mmol), and dichloromethane (4 mL) were combined and stirred at 0 °C for 10 min. After, N-(3-Dimethylaminopropyl)-N′-ethylcarbodiimide hydrochloride (76.9 mg, 0.40 mmol, 1.0 equiv.) was added and stirred at rt for 24 h. The crude product was purified by column chromatography on silica gel to obtain monoester 14 (10.6 mg, 4%). Compound 15: ^1^H NMR (500 MHz, CDCl_3_): δ 15.96 (s, 1H), 7.60 (dd, *J* = 15.8, 6.7 Hz, 2H), 7.24–7.09 (m, 7H), 7.05 (d, *J* = 1.9 Hz, 1H), 6.99 (d, *J* = 8.1 Hz, 1H), 6.94 (d, *J* = 8.2 Hz, 1H), 6.52 (dd, *J* = 26.8, 15.8 Hz, 2H), 6.05 (t, *J* = 4.5 Hz, 1H), 5.83 (s, 1H), 3.95 (s, 3H), 3.86 (s, 3H), 2.99–2.93 (m, 2H), 2.79–2.76 (m, 2H), 2.03–1.92 (m, 3H), 1.87–1.78 (m, 1H).^13^C NMR (125 MHz, CDCl_3_): δ 184.6, 181.9, 171.8, 170.5, 151.4, 148.1, 146.9, 141.3, 141.2, 139.5, 138.1, 134.4, 134.2, 129.7, 129.2, 128.3, 127.7, 126.3, 124.3, 123.4, 123.2, 121.9, 121.1, 115.0, 111.5, 109.7, 101.7, 70.7, 56.1, 56.0, 29.8, 29.2, 29.1, 18.9 ppm. IR: 3421.9, 2938.6, 2868.4, 2840.0, 1727.9, 1627.0, 1587.8, 1510.0, 1131.3 cm^−1^. MS (m/z) calcd for C_35_H_34_NaO_9_: 621.21; found: 621.4 [M+Na^+^].

### 4.2. Mice

Female and male C57BL/6 mice, 8 weeks of age, were obtained from INDICASAT’s animal facility. Mice were maintained with a 12 h light/dark cycle, at a constant temperature of 24 °C with free access to food and water. 

### 4.3. Ethics Statement

All experiments were performed in strict accordance with guidelines from the Institutional Animal Care and Use Committee and the Guide for the Care and Use of Laboratory Animals of the National Institutes of Health. The Institutional Animal Care and Use Committee of INDICASAT approved the protocol (CICUA-18-007). 

### 4.4. Macrophage Culture

Peritoneal macrophages were obtained four days after i.p. instillation of 2 mL of thioglycollate 3%, by peritoneal washing with chilled RPMI. Cells were seeded in RPMI with 10% FCS at 2 × 10^5^/well in 96-well plates and cultured for 2 h at 37 °C in an atmosphere of 5% CO_2_. Non-adherent cells were removed by washing and adherent cells were stimulated as indicated in figure legends. Cells were treated with compounds **1**–**15** (30 μM) 1 h before the stimulus with 10 ng/mL of LPS. For dose response experiments, cells were pre-treated with different concentrations (1, 3, 10, 30 μM) of compounds **1**, **2**, **4**, **5**, **6**, **8**, **10**, **13**, and **15** before the stimulation with 10 ng/mL of LPS. All the treatments and controls were performed in the presence of 0.5% of DMSO, as compounds are solubilized in this solvent. Supernatants were collected 6h after the stimulus with LPS.

### 4.5. IL-6 and PGE_2_ Measurements

Peritoneal macrophages were cultured as previously described. The concentrations of IL-6 were determined by ELISA (DuoSet kit, R&D System) according to the manufacturer’s protocol. The concentration of prostaglandin E2 (PGE_2_) was determined using the “Prostaglandin E2 ELISA Kit - Monoclonal” from Cayman CHEMICAL, according to the manufacturer’s protocol.

### 4.6. Cytotoxicity Assay

After the removal of supernatants, 100 µL of MTT (0.5 mg/mL) dissolved in RPMI were added to each well and cells were incubated overnight at 37 °C. The supernatants were removed and formazan crystals were dissolved in 100 µL of 0.04 M HCl in isopropanol. The color was analyzed at 570 nm using an ELISA plate reader. The percent of viable cells was calculated using the formula: % viability: [(OD sample) × 100%]/(OD control). The non-stimulated cells, cultured in medium plus 10% FCS and 0.5% DMSO, represented 100% viability.

### 4.7. Statistical Analysis

Results were analyzed using the statistical software package GraphPad Prism 5. Data are presented as means ± S.E.M. Statistical analysis was performed by Student’s *t*-test. A significant difference between groups was considered when *p* < 0.05. The half maximal inhibitory concentration (IC_50_) was calculated by adjusting a sigmoidal dose–response curve following GraphPad Prism5 procedure.

## 5. Conclusions

In this study, we synthesized 13 new curcumin derivatives and evaluated their effect on the production of inflammatory mediators such as TNF-α, IL-6 and PGE_2_. A common feature among these derivatives was a succinate linker that was coupled to curcumin via an esterification reaction. The alkoxide group attached to the connector varied, including aliphatic substituents to aromatic rings, where the former included acyclic structures and cyclic rings, and the latter had an increase in the presence of aromatic rings. We added the succinyl group to curcumin structure to protect the new derivatives from degradation and determine how this group affect the anti-inflammatory response. Our results indicate that a free hydroxyl group on the curcumin ring, the absence of a linker, and aromatic ligands all increase the anti-inflammatory activity of this series of compounds. In this investigation, we analyzed two different kinds of curcumin derivatives, the ether group (**2**) and succinyl group (**3**–**15**), where compound **2** continues to have the best anti-inflammatory effects of the series that we have studied. Further studies will be needed to describe the mechanism of action of this compound and to evaluate the stability and bioavailability in in vivo systems.

## Figures and Tables

**Figure 1 ijms-24-03691-f001:**
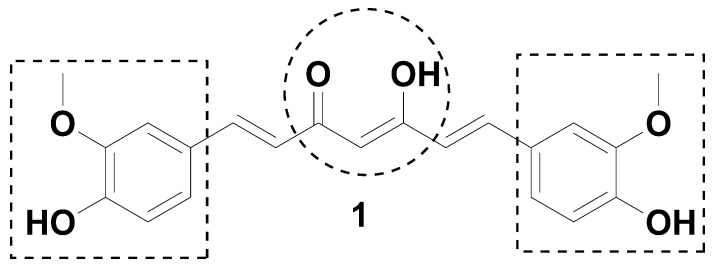
Curcumin and its reactive sites.

**Figure 2 ijms-24-03691-f002:**
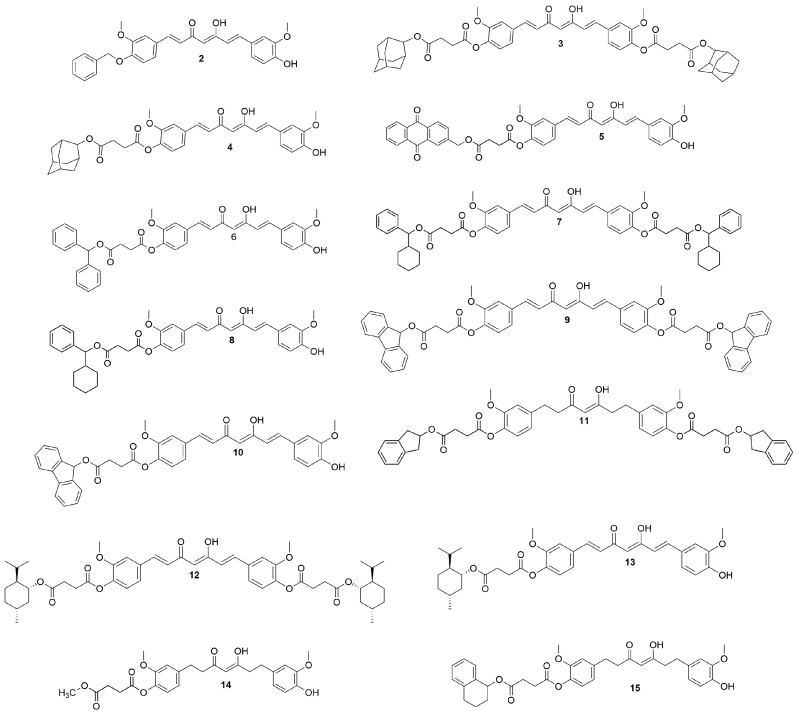
Curcumin derivatives to determine their anti-inflammatory activity.

**Figure 3 ijms-24-03691-f003:**
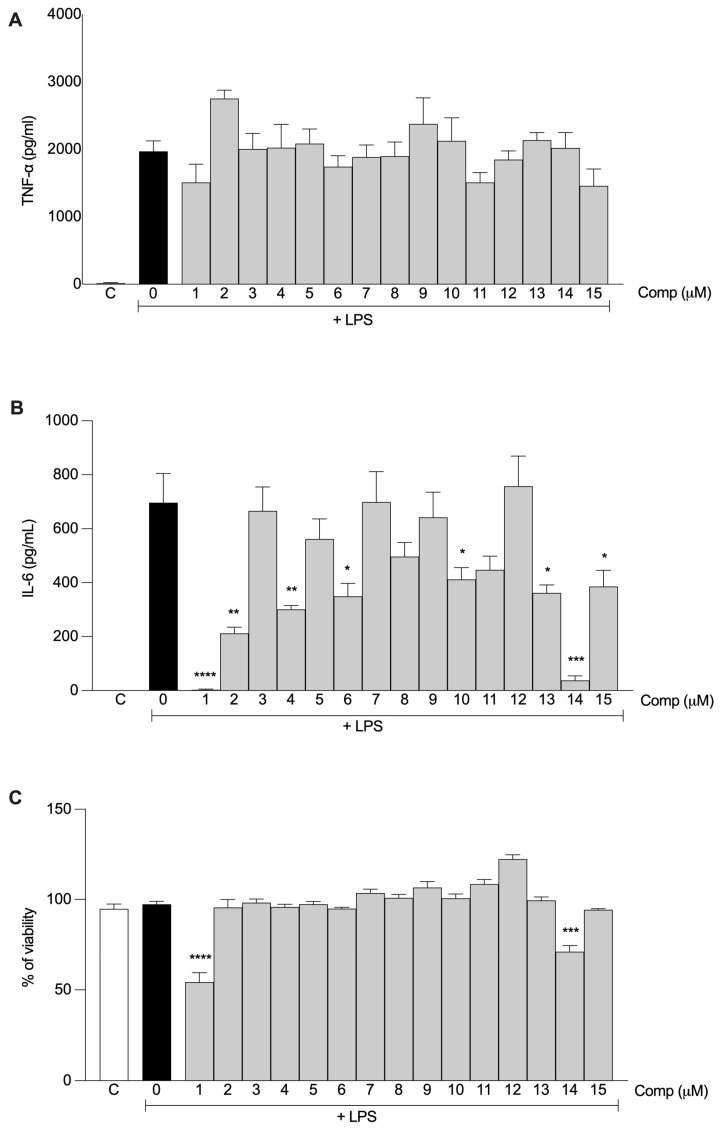
Anti-inflammatory activity of the curcumin derivatives. Peritoneal macrophages from C57BL/6 mice were pretreated with 30 μM of each compound 1 h before stimulation with 10 ng/mL LPS. After 6 h, the concentration of TNF-α (**A**) and IL-6 (**B**) in the supernatant of the cells was determined. (**C**) Cell viability was tested by MTT assays after supernatant collection. All results are presented as the mean ± S.E.M. from two independent experiments performed in triplicate. *, *p* < 0.05; **, *p* < 0.01; ***, *p* < 0.001; ****, *p* < 0.0001 relative to LPS stimulus alone (black bar). C, negative control.

**Figure 4 ijms-24-03691-f004:**
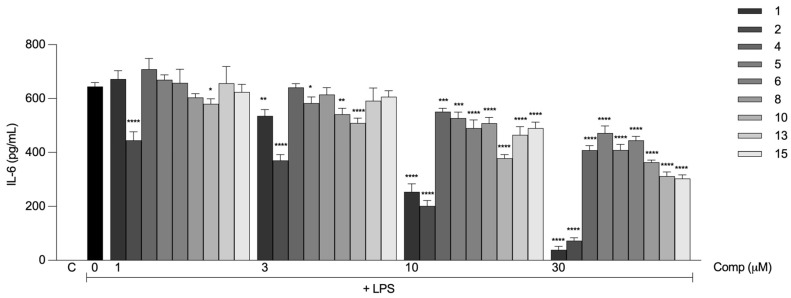
Monofunctionalized curcumin derivatives inhibit the production of IL-6 in macrophages induced by LPS. Peritoneal macrophages from C57BL/6 mice were pretreated with different concentrations (1, 3, 10, or 30 μM) of the compounds 1 h before stimulation with 10 ng/mL LPS. After 6 h, the concentrations of IL-6 in the supernatants of the cells were determined. All results are presented as the mean ± S.E.M. from three independent experiments performed in triplicate. *, *p* < 0.05; **, *p* < 0.01; ***, *p* < 0.001; ****, *p* < 0.0001 relative to LPS stimulus alone (black bar). C, negative control.

**Figure 5 ijms-24-03691-f005:**
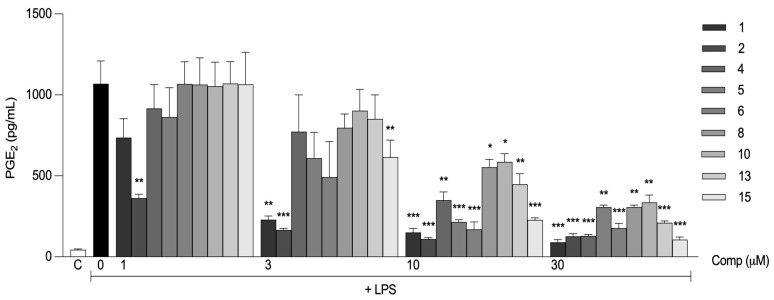
Monofunctionalized curcumin derivatives inhibit the production of PGE_2_. Peritoneal macrophages from C57BL/6 mice were pretreated with different concentrations (1, 3, 10, or 30 μM) of the compounds 1 h before stimulation with 10 ng/mL LPS. After 6 h, the concentrations of PGE_2_ in the supernatants of the cells were determined. All results are presented as the mean ± S.E.M. from two independent experiments performed in duplicate. *, *p* < 0.05; **, *p* < 0.01; ***, *p* < 0.001 relative to LPS stimulus alone (black bar). C, negative control.

**Table 1 ijms-24-03691-t001:** Chemical structures and anti-inflammatory activity of the novel curcumin derivatives.

Compound	^a^ IC_50_ ± S.D. (μM) IL-6	^a^ IC_50_ ± S.D. (μM) PGE_2_
**1**	6.85 ± 2.50	1.34 ± 0.067
**2**	3.59 ± 0.27	0.51 ± 0.08
**4**	10.5 ± 0.6	4.50 ± 3.18
**5**	7.11 ± 0.75	2.60 ± 1.75
**6**	4.21 ± 0.73	4.43 ± 0.85
**8**	1.94 ± 0.66	5.90 ± 2.38
**10**	3.60 ± 0.21	5.93 ± 2.29
**13**	10.6 ± 0.33	5.42 ± 2.04
**15**	10.4 ± 0.75	3.07 ± 0.72

^a^ Values represent the average IC_50_ from three independent experiments performed in duplicate ± S.D.

## Data Availability

Data is contained within the article and Appendix A.

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
