# Peer review of "Polyphenols with Anti-Inflammatory Properties: Synthesis and Biological Activity of Novel Curcumin Derivatives"

_ijms, 2023, doi:10.3390/ijms24043691_

Round 1
Reviewer 1 Report
This article just analyzed some compounds’ activity by LPS-stimulated macrophage. I didn’t see any rationale behind drug design. There are tons of synthesized compounds and natural products that show anti-inflammatory effects with super higher activity and safety than the compounds in the articles. Only cell studies cannot provide useful information for these compounds. The authors modified the hydroxyl group to improve the stability of curcumin, but the ester bond is easy to be hydrolyzed. Also the solubility and increased molecular weight may decrease the bioavailability of compounds.
Author Response
Reviewer
International Journal Molecular Sciences
Greetings,
We are honored that our manuscript IJMS-2122582 entitled "Polyphenols with Anti-Inflammatory Properties: Synthesis and Biological Activity of Novel Curcumin Derivatives" has been enriched with reviewer comments, and that the new changes allow it to be accepted for publication in the International Journal Molecular Sciences (IJMS) as part of a special issue "(Poly)phenols: the missing piece in the inflammation puzzle".
Based on the reviewers' comments and suggestions, the following changes were made to improve the manuscript:
Page 1 in line 4 to 18 “change of authors and affiliations was done”.
Page 1 in line 29 was added “in modulating IL-6 production”
Page 1 in line 30 was added PGE2 synthesis.
Page 1 in line 38 was corrected “pleiotropic molecule”
Page 2 in line 43 was changed at this position the paragraph “Curcumin displays anti-inflammatory effects by modulating several….. and plasma has prevented the medical use of curcumin [23].”
Page 2 in line 61 was added “sites of curcumin “
Page 2 in line 68 to 72 was added “In 2011, Wichitnithad et.al demonstrated that succinylation of curcuminoids protects the curcumin from hydrolysis and is an effective strategy against colon cancer [5]. Hence, we added the succinyl group to curcumin structure to protect the new molecules from degradation and to improve their bioavailability”
Page 2 in line 76 was added “In… “
Reviewer 1: Improving the introduction and design of experiment: Page 2 in line 77 to 80 was added “Specifically, we focus on protecting the hydroxyl groups of the aromatic ring of curcumin through the incorporation of the succinyl group. We evaluated how the anti-inflammatory activity of the derivatives was altered as a result of the structural changes compared with curcumin”
Page 4 in line 174 was added “IL-6 and TNF-α.”
Page 4 in line 176 was added “None of the compounds influenced TNF-α production (data not shown). “
Page 4 in line 177 was added “Compound”.
Page 5 in line 194 was added “Figure 4”
Page 6 in line 207 to 210 was added “its synthesis is influenced by the enzyme COX-2 [29]. It has been shown that curcumin can suppress COX-2 expression and PGE2 production in models of inflammation [30]. We decided to examine whether these curcumin derivatives preserved the effect on PGE2 production.”
Page 7 in line 236 was added “with statistically significant”
Page 7 in line 250 to 251 was added “aromatic rings in the benzylic position. Among the compounds with two aromatic rings attaching to the benzylic position, 6 and 10 showed the greater effect on IL-6 production.
Page 8 in line 258 to 260 was added “(i) the phenolic ring of the unesterified curcumin derivatives can form hydrogen bonds with another molecule, affecting the activity of these compounds; (ii) a π-π interaction with the curcumin ligand is possible since compounds 2, 6, 8, and 10 showed “
Page 8 in line 274 was added “those found with IL-6”
Page 8 in line 279 to 282 was added “PGE2 by a mechanism that involves regulation of COX-2 activity [31]. Since curcumin derivatives presented herein affect the production of PGE2, further studies are necessary to elucidate a mechanism engaged in this effect.”
Page 12 in line 432 was added “found: 859.6 [M+Na+]”.
Page 12 in line 439 was added “C35H38NaO9: 625.24; found: 625.4 [M+Na+].”
Page 12 in line 454 to 455 was added “MS (m/z) calcd for C40H32O11: 688.6; found: 689.4 [M+H+].
Page 12 in line 470 to 471 was added “MS (m/z) calcd for C38H34NaO9: 657.21; found: 657.4 [M+Na+].”
Page 13 in line 489 to 490 was added “MS (m/z) calcd for C55H60NaO12: 935.40; found: 935.8 [M+Na+].”
Page 13 in line 497 was added “MS (m/z) calcd for C38H40NaO9: 663.26; found: 663.5 [M+Na+].”
Page 14 in line 531 was added “MS (m/z) calcd for C47H44NaO12: 823.27; found: 823.5 [M+Na+].“
Page 15 in line 574 to 575 was added “MS (m/z) calcd for C35H34NaO9: 621.21; found: 621.4 [M+Na+].”
Page 15 in line 591 to 592 was added “MS (m/z) calcd for C35H34NaO9: 621.21; found: 621.4 [M+Na+].”
Page 16 in line 639 to 641 was added “We added the succinyl group to curcumin structure to protect the new derivatives from degradation and determine how this group affect the anti-inflammatory response.“
Page 16 in line 643 was added “anti-inflammatory activity”
Page 19 in line 752 to 753 the reference # 30 was added “Curcumin inhibits cyclooxygenase-2 transcription in bile acid- and phorbol ester-treated human gastrointestinal epithelial cells.”
We look forward to hearing from you,
Best regards
Johant Lakey Beitia, M.Sc, Ph.D
Center for Biodiversity and Drug Discovery
jlakey@indicasat; Tel: +(507) 517-0700
https://orcid.org/0000-0001-7554-4043

Reviewer 2 Report
The manuscript by Lakey-Beitia et al reports on the synthesis of several derivatives of curcumin and the validation of their antiinflammatory properties in vitro. The work is well designed and presented and its results relevant to the eventual usage of improved curcuminversions as antiinflammatory drugs.
The paper has some minor issues to be addressed by the authors.
Introduction, page 1 line 39: "Pleotropic" should be "pleiotropic"
Discussion, page 7, line 222: Authors state that "the curcumin derivatives inhibited the produciotn of IL-6 and PGE2 wihout affecting TNFa secretion". However, in their submitted work no measurement of TNFa is presented.
Author Response
Reviewer
International Journal Molecular Sciences
Greetings,
We are honored that our manuscript IJMS-2122582 entitled "Polyphenols with Anti-Inflammatory Properties: Synthesis and Biological Activity of Novel Curcumin Derivatives" has been enriched with reviewer comments, and that the new changes allow it to be accepted for publication in the International Journal Molecular Sciences (IJMS) as part of a special issue "(Poly)phenols: the missing piece in the inflammation puzzle".
Based on the reviewers' comments and suggestions, the following changes were made to improve the manuscript:
Page 1 in line 38 was corrected “pleiotropic molecule”
Page 4 in line 174 was added “IL-6 and TNF-α.”
Page 4 in line 176 was added “None of the compounds influenced TNF-α production (data not shown). “
Page 16 in line 639 to 641 was added “We added the succinyl group to curcumin structure to protect the new derivatives from degradation and determine how this group affect the anti-inflammatory response.“
Page 16 in line 643 was added “anti-inflammatory activity”
We look forward to hearing from you,
Best regards
Johant Lakey Beitia, M.Sc, Ph.D
Center for Biodiversity and Drug Discovery
jlakey@indicasat; Tel: +(507) 517-0700
https://orcid.org/0000-0001-7554-4043
